# A High-Order Kalman Filter Method for Fusion Estimation of Motion Trajectories of Multi-Robot Formation

**DOI:** 10.3390/s22155590

**Published:** 2022-07-26

**Authors:** Miao Wang, Weifeng Liu, Chenglin Wen

**Affiliations:** 1School of Electrical and Control Engineering, Shaanxi University of Science and Technology, Xi’an 710021, China; 200611024@sust.edu.cn (M.W.); liuwf@sust.edu.cn (W.L.); 2School of Automation, Guangdong University of Petrochemical Technology, Maoming 525000, China

**Keywords:** multi-robot formation, trajectory estimation, higher-order Kalman filter, fusion estimation

## Abstract

Multi-robot motion and observation generally have nonlinear characteristics; in response to the problem that the existing extended Kalman filter (EKF) algorithm used in robot position estimation only considers first-order expansion and ignores the higher-order information, this paper proposes a multi-robot formation trajectory based on the high-order Kalman filter method. The joint estimation method uses Taylor expansion of the state equation and observation equation and introduces remainder variables on this basis, which effectively improves the estimation accuracy. In addition, the truncation error and rounding error of the filtering algorithm before and after the introduction of remainder variables, respectively, are compared. Our analysis shows that the rounding error is much smaller than the truncation error, and the nonlinear estimation performance is greatly improved.

## 1. Introduction

As a typical complex system, swarm dynamics systems have been widely studied since first being was proposed. A swarm is composed of many simple intelligent individuals. The interaction between individuals based on basic mechanical rules can stimulate highly coordinated swarm behavior [1,2,3,4]. The dynamic behavior of swarming in nature shows amazing charm. The system composed of interconnected and constantly moving individuals emerges as colorful and highly coordinated swarming behavior, which provides a rich source of ideas for the cognition and optimization of industrial and social groups. It has considerable prospects in the fields of multi-agent collaboration [5], UAV formation [6], multi-robot collaboration [7], and intelligent grids [8].

In a multi-robot formation system, because the robot can detect the external environment it is a control system with pattern recognition, complex task execution and allocation, autonomous behavior decision-making, and other functions necessary to operate in a hostile environment and complete a variety of complex tasks. Therefore, a multi-robot formation system exhibits the characteristics of high consistency and collaboration with the outside world. In this relatively complex control system, it is necessary to obtain its precise state quantity first in order for control to be applied and to achieve the purpose of completing complex tasks. Therefore, it is necessary to perform more accurate state estimation for multi-robot motion.

Among the state estimation methods, in 1960 R. E. Kalman et al. [9,10] proposed a recursive state estimation method suitable for a system in which the model is linear and the noise obeys a Gaussian distribution. Kalman filtering (KF)is a time domain filtering method; when the system process noise and measurement noise are mutually independent Gaussian white noise with zero mean, a Kalman filter is the optimal unbiased estimator in the sense of minimum variance [11].

In general, the motion equation of a multi-robot formation and the measurement equation of its sensor are both nonlinear equations. The motion equation has weak nonlinear characteristics because only simple operations such as turning and going straight are available. The sensor observes the robot target, which shows strong nonlinear characteristics. For nonlinear state estimation problems, the traditional filtering methods are mainly extended Kalmanl filter (EKF), unscented Kalman filter (UKF) and cubature Kalman filter (CKF). EKF uses a Taylor series first-order approximation expansion of the nonlinear equations, which introduces linearization errors and has low estimation accuracy for estimating systems with strong nonlinearity [12]. Over time, Julier et al. [13] proposed another nonlinear filtering method, the unscented Kalman filter. The core of the UKF algorithm is the UT transformation, which uses an ensemble of sigma sampling points to approximate the nonlinear system model [14,15,16]. The UKF algorithm is more computationally intensive than the EKF algorithm, and its filtering accuracy is better, at least to the second-order accuracy of the Taylor expansion. However, the shortcomings of the UKF algorithm are obvious in that it can only be applied to Gaussian systems, and a problem with the non-positive definite error covariance matrix is caused by the emergence of negative weight in the implementation of the algorithm [17]. The frequency of this problem becomes greater as the order of the system increases and the nonlinearity grows stronger. In order to better solve the problem of UKF being non-positive definite when dealing with high-dimensional covariance matrices, resulting in divergent filtering results, Ienkaram proposed a weight-selective UKF filtering method called Cubature Kalman Filter (CKF) [18,19]. CKF optimizes the sigma point sampling method and weight distribution in UKF using a spherical integral and radial integral. Compared with traditional EKF, it has improved estimation accuracy and improved stability of filtering methods [20].

In recent years, in the treatment of nonlinear estimation problems, the structure of the extended Kalman filter (EKF) is used to design recursive estimators as a new idea to solve the problem.The main idea is to transform the linear error of the nonlinear equation after doing the first-order Taylor expansion into the form of a product of the scaling matrix associated with the problem solved and some time-varying positive definite matrix, then set the upper bound of the covariance matrix [21,22,23], and get the upper bound of the minimum covariance matrix by numerically solving the Ricatti equation, and finally use recursive filtering, which cannot be accurately resolved to get this scaling matrix, and then the process of solving the inverse matrix in the covariance matrix obtained by the Ricatti equation is more complicated, which greatly increases the computational burden in some high-dimensional occasions and cannot meet the real-time requirements [24]. The method cannot accurately analyze the scaling matrix, and the process of solving the inverse matrix in the covariance matrix obtained by the Ricatti equation is relatively complicated. The other is to borrow the Taylor expansion idea, expand the higher-order term with the original equation, design the new state and measurement equation, and use recursive filtering to get the required state estimate [25,26]. It can choose the expansion order reasonably according to the nonlinear strength and weakness of the actual problem, which is a good balance between the requirements of arithmetic power and estimation accuracy, and has a good real-time filtering feature [27,28]. At the same time, this method has also been extended to correlation entropy filtering [29], Kronecker product [30] and lithium battery life prediction [31], which all show good estimation performance.

In traditional state estimation for multi-robot operation, the nonlinear state estimation is performed separately for each robot using methods such as EKF, UKF, or CKF. In addition to the problems already mentioned in this paper, performing individual state estimation for each robot reduces the estimation accuracy [32,33,34]. In recent years, the use of trigger probability in the multi-robot formation problem as the connection relationship between robots has gradually become a mainstream idea [24]. The reason for real-time estimation is that at a certain moment the connection relationship between multiple robots is deterministic, and there is only a connection or no connection, which is equivalent to a binary connection.

On this basis, this paper addresses the multi-robot formation state estimation problem by proposing a centerless multi-robot state joint estimation method based on high-order extended Kalman filter (HEKF). This improves the original multi-robot connection relationship based on trigger probability and improves the multi-robot state estimation accuracy. The main algorithm of this paper is as follows. In view of the nonlinear characteristics of multi-robot motion and observation, Taylor’s higher-order expansion of the robot’s state equation and measurement equation is performed to solve the problem of different nonlinear degrees of the two nonlinear equations. The high-order Kalman filtering algorithm is improved by expanding the truncation error with the state for dimensional estimation and fusing the state information of neighboring node robots to improve the multi-robot state estimation accuracy. A numerical simulation experiment proves the effectiveness of the algorithm. Therefore, we choose the second method based on high-order Kalman filtering. The main work of this paper includes:After Taylor expansion of the nonlinear state equation and observation equation, the remainder variable is introduced and the original EKF algorithm and the high-order Kalman filtering algorithm are changed to discard the truncation error. Only the first-order and low-order items are retained, reducing the estimation error and improving the estimation accuracy.A dynamic model of the remainder variables is introduced into the state equation and observation equation as hidden variables, and the changed pseudo-linear state equation and observation equation are rewritten into higher-order linear forms through dimension expansion.The new high-order linear equation is used to obtain the state estimation value using the recursive filtering algorithm, and the effectiveness of the algorithm is analyzed.

The rest of this paper is organized as follows. Section 2 proposes state equations and observation equations for the multi-robot formation operating system and designs a recursive estimator based on the EKF structure, while Section 3 presents the recursive estimator form derived from the second section, introduces the remainder variables into the state equation and the observation equation, and presents a new high-order linear filter derived through dynamic modeling, then analyzes its performance. In Section 4, an indoor multi-robot simulation numerical experiment is used to verify the effectiveness of the algorithm proposed in this paper. Finally, conclusions are provided in Section 5. In addition, the important mathematical symbols in Appendix A.

## 2. Problem Description

The movement of the robot is generally carried out by operations such as going straight and turning, and the observation system is generally composed of radar or visual sensors. Therefore, both the robot’s motion and the observation system have nonlinear characteristics. In a multi-robot formation environment, there is a certain degree of motion consistency between the robots, and thus a certain coupling relationship. The dynamic equations can be described by the following *N* node coupling equations:(1)$$x_{i}\left(k+1\right)=f\left(x_{i}\left(k\right)\right)+c\sum_{j=1}^{N}\Gamma x_{j}\left(k\right)+w_{i}\left(k\right)$$
(2)$$y_{i}\left(k+1\right)=h\left(x_{i}\left(k+1\right)\right)+v_{i}\left(k+1\right)$$
where $x_{i}\left(k\right)\in R^{n}$ and $y_{i}\left(k+1\right)\in R^{q}$ denote the state vector and the measurement vector, respectively, *i* and *k* denote the node index and the time instant, respectively, f(·) and h(·) are known nonlinear functions that are assumed to be continuously differentiable, c>0 is the overall coupling strength, and Γ is the inner-coupling matrix. The process noise, $w_{i}\left(k\right)$,and the measurement noise, $v_{i}\left(k+1\right)$, are assumed to be mutually uncorrelated zero-mean white Gaussian noise with the covariances $Q_{i}\left(k\right)$ and $R_{i}\left(k+1\right)$, respectively.

The EKF structure is used to design the recursive estimation of the multi-robot formation trajectories (1) and (2):(3)$${\hat{x}}_{i}\left(k+1|k\right)=f\left({\hat{x}}_{i}\left(k|k\right)\right)+c\sum_{j=1}^{N}\Gamma {\hat{x}}_{j}\left(k|k\right)$$
(4)$${\hat{x}}_{i}\left(k+1|k+1\right)={\hat{x}}_{i}\left(k+1|k\right)+K_{i}\left(k+1\right)\left[y_{i}\left(k+1\right)-h\left({\hat{x}}_{i}\left(k+1|k\right)\right)\right]$$
where ${\hat{x}}_{i}\left(k+1|k\right)$ and ${\hat{x}}_{i}\left(k+1|k+1\right)$ denote the predicted and updated estimates at time instant k+1, respectively., and $K_{i,k+1}$ is the gain matrix to be determined

As in the EKF, we define the updated estimation error and the corresponding covariance as follows:(5)$$e_{i}\left(k+1|k+1\right)=x_{i}\left(k+1\right)-{\hat{x}}_{i}\left(k+1|k+1\right)$$
(6)$$P_{i}\left(k+1|k+1\right)=E\left\{\left.e_{i}\left(k+1|k+1\right)e_{i}^{T}\left(k+1|k+1\right)\right\}\right.$$

## 3. Algorithm Description

In the second section, we use the EKF structure to design the recursive filter. This idea generally expands the nonlinear function to obtain the first-order terms and truncation errors. The traditional EKF algorithm directly discards the truncation errors and keeps only the first-order terms. In the transfer function with strong non-linearity a large amount of information will be lost, which makes the estimation result inaccurate. In recent years, in the study of nonlinear filtering problems, the following two ideas have mainly been adopted in the design of recursive filters using the EKF structure. The first is to transform the truncation error into the form of a product of the scaling matrix and a time-varying positive definite matrix related to the problem, then set the upper bound of the covariance matrix, obtain the upper bound of the minimum covariance matrix by numerically solving the Ricatti equation, and finally use recursive filtering, which cannot accurately analyze the scaling matrix, before obtaining the covariance matrix through the Ricatti equation. The inverse matrix process is more complicated, and in occasions with high dimensions the computational burden is greatly increased to the point that real-time requirements cannot be met. The second is to use the idea of Taylor expansion to expand the high-order term while using the original equation to expand the dimension, design new state and measurement equations, use recursive filtering to obtain the required state estimation value, and avoid first above method. The “dimension disaster” caused by inversion in the dimensional situation, as well as the expansion order, can be reasonably selected according to the nonlinear strong and weak characteristics in the actual problem, which takes into account the requirements of computing power and estimation accuracy and has a good real-time filtering feature. A block diagram of the algorithm is shown in Figure 1.

### 3.1. Taylor Expansion of Nonlinear System and Introduction of Remainder Variables

Among the nonlinear filtering algorithms, the extended Kalman filter (EKF) algorithm is widely used. Its main idea is perform a first-order Taylor expansion of the nonlinear state equation around the filter estimated value. The main process is as follows.

Let $f_{i}\left(k\right)=f\left(x_{i}\left(k\right)\right)+c\sum_{j=1}^{N}\Gamma x_{j}\left(k\right)$; then, $x_{i}\left(k+1\right)$ is subjected to a first-order Taylor expansion at the filter estimate ${\hat{x}}_{i}\left(k|k\right)$ to obtain the following expression for $x_{i}\left(k+1\right)$: (7)$$\begin{array}{cc} x_{i}\left(k+1\right) & =f_{i}\left(k\right)+w_{i}\left(k\right)\\ \; & =f_{i}\left({\hat{x}}_{i}\left(k|k\right)\right)+{\left.\frac{\partial f_{i}}{\partial x}\right|}_{x_{i}\left(k\right)={\hat{x}}_{i}\left(k|k\right)}\left(x_{i}\left(k\right)-{\hat{x}}_{i}\left(k|k\right)\right)+T\left(f_{i}\left({\hat{x}}_{i}\left(k|k\right)\right)\right)+w_{i}\left(k\right)\\ \; & =A_{i}\left({\hat{x}}_{i}\left(k|k\right),k\right)x_{i}\left(k\right)+\beta_{i}\left({\hat{x}}_{i}\left(k|k\right),\xi_{i}\left(k\right)\right)+w_{i}\left(k\right)\\ & =A_{i}\left({\hat{x}}_{i}\left(k|k\right),k\right)x_{i}\left(k\right)+f_{i}\left({\hat{x}}_{i}\left(k|k\right)\right)-A_{i}\left({\hat{x}}_{i}\left(k|k\right),k\right){\hat{x}}_{i}\left(k|k\right)+T\left(f\left({\hat{x}}_{i}\left(k|k\right)\right)\right)+w_{i}\left(k\right) \end{array}$$
where the state transition matrix is ${\left.A_{i}\left({\hat{x}}_{i}\left(k|k\right),k\right)=\frac{\partial f_{i}}{\partial x}\right|}_{x_{i}\left(k\right)={\hat{x}}_{i}\left(k|k\right)}$, and $\beta_{i}\left({\hat{x}}_{i}\left(k|k\right),\xi_{i}\left(k\right)\right)=f_{i}\left({\hat{x}}_{i}\left(k|k\right)\right)-A_{i}\left({\hat{x}}_{i}\left(k|k\right)\right){\hat{x}}_{i}\left(k|k\right)+T(f_{i}\left({\hat{x}}_{i}\left(k|k\right)\right)$ is the state remainder variables.

In the same way, the first-order Taylor expansion of the nonlinear observation function $h_{i}\left(x_{i}\left(k+1\right)\right)$ at the state prediction value ${\hat{x}}_{i}\left(k+1|k\right)$ can be obtained:(8)$$\begin{array}{cc} y_{i}\left(k+1\right) & =h\left(x_{i}\left(k+1\right)\right)+v_{i}\left(k+1\right)\\ \; & =h\left({\hat{x}}_{i}\left(k+1|k\right)\right)+{\left.\frac{\partial h_{i}}{\partial x}\right|}_{x_{i}\left(k+1\right)={\hat{x}}_{i}\left(k+1|k\right)}\left(x_{i}\left(k+1\right)-{\hat{x}}_{i}\left(k+1|k\right)\right)\\ \; & +T\left(h\left({\hat{x}}_{i}\left(k+1|k\right)\right)\right)+v_{i}\left(k+1\right)\\ \; & =h\left({\hat{x}}_{i}\left(k+1|k\right)\right)+C_{i}\left({\hat{x}}_{i}\left(k+1|k\right),k+1\right)\left(x_{i}\left(k+1\right)-{\hat{x}}_{i}\left(k+1|k\right)\right)\\ \; & +T\left({\hat{x}}_{i}\left(k+1|k\right)\right))+v_{i}\left(k+1\right)\\ \; & =C_{i}\left({\hat{x}}_{i}\left(k+1|k\right),k+1\right)x_{i}\left(k+1\right)+\gamma \left({\hat{x}}_{i}\left(k+1|k\right)\right)+v_{i}\left(k+1\right) \end{array}$$
where the observation matrix is ${\left.C_{i}\left({\hat{x}}_{i}\left(k+1|k\right),k+1\right)=\frac{\partial h_{i}}{\partial x}\right|}_{x_{i}\left(k+1\right)={\hat{x}}_{i}\left(k+1|k\right)}$, and $\gamma_{i}\left({\hat{x}}_{i}\left(k+1|k\right)\right)=y_{i}\left(k+1\right)-h\left({\hat{x}}_{i}\left(k+1|k\right)\right)-C_{i}\left({\hat{x}}_{i}\left(k+1|k\right),k+1\right){\hat{x}}_{i}\left(k+1|k\right)+T\left(h\left({\hat{x}}_{i}\left(k+1|k\right)\right)\right)$ is the observation remainder.

From this, we can see that in the original extended Kalman filtering algorithm, which uses only the first-order term information of the nonlinear function, the high-order term information and the truncation error are directly discarded, which makes the estimation accuracy low in occasions with a high degree of nonlinearity and leads to inaccurate estimation results. In certain extreme cases, the filtering results may even directly diverge. Here, we introduce remainder variables into the state equation and observation equation, respectively, to compensate for the loss of high-order information in the original filtering algorithm.

### 3.2. Dynamic Modeling of Remainder Variables and Establishment of Higher-Order Linear Systems

The remainder variable is obtained by integrating the constant term with the truncation error after Taylor expansion of the nonlinear state function and the observation function around the filter estimation value and the one-step prediction value, respectively. By using the remainder variable information to improve the estimation accuracy, the original state equation and observation equation need to be expanded and rewritten to establish a high-order Kalman filter.

**Lemma** **1.**
*The n-order Taylor formula of the binary function z=f(x,y) at $\left(x_{0},y_{0}\right)$:*

*Assuming that z=f(x,y) is continuous in a neighborhood of $\left(x_{0},y_{0}\right)$ and has continuous partial derivatives up to order; if $\left(x_{0}+h,y_{0}+h\right)$ is any point in this neighborhood, there are*

(9)
$$\begin{array}{cc} \; & f\left(x_{0}+h,y_{0}+k\right)=f\left(x_{0},y_{0}\right)+\left(h\frac{\partial }{\partial x}+k\frac{\partial }{\partial y}\right)f\left(x_{0},y_{0}\right)+\frac{1}{2!}{\left(h\frac{\partial }{\partial x}+k\frac{\partial }{\partial y}\right)}^{2}f\left(x_{0},y_{0}\right)\\ \; & +\cdots +\frac{1}{n!}{\left(h\frac{\partial }{\partial x}+k\frac{\partial }{\partial y}\right)}^{n}f\left(x_{0},y_{0}\right)+\frac{1}{(n+1)!}{\left(h\frac{\partial }{\partial x}+k\frac{\partial }{\partial y}\right)}^{n+1}f\left(x_{0}+\theta h,y_{0}+\theta k\right)\\ \; & \left(0<\theta <1\right) \end{array}$$


*where*

(10)
$${\left(h\frac{\partial }{\partial x}+k\frac{\partial }{\partial y}\right)}^{m}f\left(x_{0},y_{0}\right)=\sum_{p=0}^{m}C_{m}^{p}h^{p}k^{m-p}\frac{\partial^{m}f\left(x_{0},y_{0}\right)}{\partial x^{p}\partial y^{m-p}}$$



**Lemma** **2.**
*Taylor expansion of a multivariate function at $x_{k}$:*

(11)
$$\begin{array}{cc} \; & f\left(x^{1},x^{2},\cdots ,x^{n}\right)=f\left(x_{k}^{1},x_{k}^{2},\cdots ,x_{k}^{n}\right)+\sum_{i=1}^{n}\left(x^{i}-x_{k}^{i}\right){f^{\prime }}_{x^{i}}\left(x_{k}^{1},x_{k}^{2},\cdots ,x_{k}^{n}\right)\\ \; & +\frac{1}{2!}\sum_{i=1}^{n}\sum_{j=1}^{n}\left(x^{i}-x_{k}^{i}\right)\left(x^{j}-x_{k}^{j}\right){f^{\prime \prime }}_{ij}\left(x_{k}^{1},x_{k}^{2},\cdots ,x_{k}^{n}\right)+o\left(f\right) \end{array}$$


*Assume that $X={\left[x^{1},x^{2},\cdots ,x^{n}\right]}^{T}$*

(12)
$$f\left(X\right)=f\left(X_{k}\right)+{\left[\nabla f\left(X_{k}\right)\right]}^{T}\left(X-X_{k}\right)+\frac{1}{2!}{\left[X-X_{k}\right]}^{T}H\left(X_{k}\right)\left[X-X_{k}\right]+T\left(f\right)$$


*where*

(13)
$$\nabla f\left(X_{k}\right)={\left[\frac{\partial f(X_{k})}{\partial x^{1}},\frac{\partial f(X_{k})}{\partial x^{2}},\cdots ,\frac{\partial f(X_{k})}{\partial x^{n}}\right]}^{T}$$


(14)
$$H\left(X_{k}\right)=\left[\begin{array}{cccc} \frac{\partial^{2}f\left(X_{k}\right)}{\partial {x^{1}}^{2}} & \frac{\partial^{2}f\left(X_{k}\right)}{\partial x^{1}\partial x^{2}} & \cdots & \frac{\partial^{2}f\left(X_{k}\right)}{\partial x^{1}\partial x^{n}}\\ \frac{\partial^{2}f\left(X_{k}\right)}{\partial x^{2}\partial x^{1}} & \frac{\partial^{2}f\left(X_{k}\right)}{\partial {x^{2}}^{2}} & \cdots & \frac{\partial^{2}f\left(X_{k}\right)}{\partial x^{2}\partial x^{n}}\\ \vdots & \vdots & \ddots & \vdots \\ \frac{\partial^{2}f\left(X_{k}\right)}{\partial x^{n}\partial x^{1}} & \frac{\partial^{2}f\left(X_{k}\right)}{\partial x^{n}\partial x^{2}} & \cdots & \frac{\partial^{2}f\left(X_{k}\right)}{\partial {x^{n}}^{2}} \end{array}\right]$$



#### 3.2.1. Pseudo-Linear Representation of State Function $f_{i}\left(x_{i,k}\right)$

According to the above lemma, the state transition function $f_{i}\left(x_{i,k}\right)$ in Formula (1) can be expressed as a polynomial form using Taylor expansion, as follows.

For simplicity, the following derivation takes two state variables, $x_{1}$ and $x_{2}$:(15)$$\begin{array}{cc} f_{i}\left(x\left(k\right)\right) & =a_{i,0}+\left(a_{i,1,0}x_{1}\left(k\right)+a_{i,0,1}x_{2}\left(k\right)\right)\\ \; & +(a_{i,2,0}x_{1}^{2}\left(k\right)+a_{i,1,1}x_{1}\left(k\right)x_{2}\left(k\right)+a_{i,0,2}x_{2}^{2}\left(k\right))\\ \; & +(a_{i,3,0}x_{1}^{3}\left(k\right)+a_{i,2,1}x_{1}^{2}\left(k\right)x_{2}\left(k\right)+a_{i,1,2}x_{1}\left(k\right)x_{2}^{2}\left(k\right)+a_{i,3,0}x_{2}^{3}\left(k\right))\\ \; & +\cdots +\sum_{\begin{array}{c} l_{1}+l_{2}=l\\ l_{1,}l_{2}<l \end{array}}a_{i,l_{1},l_{2}}x_{1}^{l_{1}}\left(k\right)x_{2}^{l_{2}}\left(k\right)\\ \; & +\cdots +\sum_{\begin{array}{c} r_{1}+r_{2}=r\\ r_{1,}r_{2}<r \end{array}}a_{i,r_{1},r_{2}}x_{1}^{r_{1}}\left(k\right)x_{2}^{r_{2}}\left(k\right)\\ \; & +\beta \left(x_{1}\left(k\right)\right)+\beta \left(x_{2}\left(k\right)\right) \end{array}$$
where $a_{i,0}$ is a constant, $\sum_{\begin{array}{c} l_{1}+l_{2}=l\\ l_{1,}l_{2}<l \end{array}}a_{i,l_{1},l_{2}}x_{1}^{l_{1}}\left(k\right)x_{2}^{l_{2}}\left(k\right)$, $0<l_{1},l_{2}<l\left(l=1,2,\cdots ,r\right)$ are the weighted sum of all *l* order tensor terms, $a_{i,l_{1},l_{2}},l_{1}+l_{2}=l,0\leq l_{1},l_{2}\leq l\left(l=1,2,\cdots ,r\right)$ is the weight of the corresponding term, and $\beta \left(x_{1}\left(k\right)\right),\beta \left(x_{2}\left(k\right)\right)$ are the respective remainder variables of the state.

**Theorem** **1.**
*Let $x^{l}\left(k\right)=x_{1}^{l_{1}}\left(k\right)x_{2}^{l_{2}}\left(k\right),l_{1}+l_{2}=l,0<l_{j}<l\left(l=1,2,\cdots ,r\right)$ be a set of l-order hidden variables corresponding to the original system variable x(k).*


**Theorem** **2.**
*The weight vector corresponding to the l-order latent variable vector is then*

(16)
$$a_{i}^{\left(l\right)}=\left[\begin{array}{cccc} a_{i;1}^{\left(l\right)} & a_{i;2}^{\left(l\right)} & \cdots & a_{i;n_{l}}^{\left(l\right)} \end{array}\right]=\left[\begin{array}{cccc} a_{i;l,0} & a_{i;l-1,1} & \cdots & a_{i;0,l} \end{array}\right]\left(i=1,2\right)$$



We use the following two states as an example: (17)$$\left[\begin{array}{c} x_{1}^{\left(1\right)}\left(k+1\right)\\ x_{2}^{\left(1\right)}\left(k+1\right) \end{array}\right]=\left[\begin{array}{cccccc} a_{1}^{\left(1\right)} & a_{1}^{\left(2\right)} & \cdots & a_{1}^{\left(l\right)} & \cdots & a_{1}^{\left(r\right)}\;\begin{array}{cc} a_{1}\left(\beta \left(x{}_{1}\right)\right) & a_{1}\left(\beta \left(x{}_{2}\right)\right) \end{array}\\ a_{2}^{\left(1\right)} & a_{2}^{\left(2\right)} & \cdots & a_{2}^{\left(l\right)} & \cdots & a_{2}^{\left(r\right)}\;\begin{array}{cc} a_{2}\left(\beta \left(x{}_{1}\right)\right) & a_{2}\left(\beta \left(x{}_{2}\right)\right) \end{array} \end{array}\right]\left[\begin{array}{c} x^{\left(1\right)}\left(k\right)\\ x^{\left(2\right)}\left(k\right)\\ \vdots \\ x^{\left(l\right)}\left(k\right)\\ \vdots \\ x^{\left(r\right)}\left(k\right)\\ \beta (x_{1}\left(k\right))\\ \beta (x_{2}\left(k\right)) \end{array}\right]+\left[\begin{array}{c} w_{1}^{\left(1\right)}\left(k\right)\\ w_{2}^{\left(1\right)}\left(k\right) \end{array}\right]$$
Donate $x\left(k\right)=x^{\left(1\right)}\left(k\right)=\left[\begin{array}{c} x_{1}^{\left(1\right)}\left(k\right)\\ x_{2}^{\left(1\right)}\left(k\right) \end{array}\right],A^{\left(l\right)}=\left[\begin{array}{c} a_{1}^{\left(l\right)}\\ a_{2}^{\left(l\right)} \end{array}\right],w\left(k\right)=\left[\begin{array}{c} w_{1}^{\left(1\right)}\left(k\right)\\ w_{2}^{\left(1\right)}\left(k\right) \end{array}\right]$

Then,
(18)$$\begin{array}{cc} x^{\left(1\right)}\left(k+1\right) & =\left[\begin{array}{cccccc} A^{\left(1\right)} & A^{\left(2\right)} & \cdots & A^{\left(l\right)} & \cdots & \begin{array}{cc} A^{\left(r\right)} & A(\beta ) \end{array} \end{array}\right]\left[\begin{array}{c} x^{\left(1\right)}\left(k\right)\\ x^{\left(2\right)}\left(k\right)\\ \vdots \\ x^{\left(l\right)}\left(k\right)\\ \vdots \\ x^{\left(r\right)}\left(k\right)\\ \beta (x(k)) \end{array}\right]\;+w^{\left(1\right)}\left(k\right)\\ \; & =A^{\left(1\right)}x^{\left(1\right)}\left(k\right)+\sum_{l=2}^{r}A^{\left(l\right)}x^{\left(l\right)}\left(k\right)+A\left(\beta \right)\beta \left(x\left(k\right)\right)+w^{\left(l\right)}\left(k\right) \end{array}$$

**Remark** **1.**
*Compared with the original state model, $x^{\left(1\right)}\left(k\right)={\left[\begin{array}{cc} x_{1}\left(k\right) & x_{2}\left(k\right) \end{array}\right]}^{T}$, the above equation is in a pseudo-linear form with higher-order latent variables $x^{\left(l\right)}\left(k\right)\left(l=1,2,\cdots ,r\right)$ and remainder variables β(x(k)). There is only a change in representation; as there is no essential difference, this is called pseudo-linearization.*


#### 3.2.2. Pseudolinear Representation of the Measurement Function

Similarly, the measurement function in formula (2) is expressed as a polynomial form by Taylor expansion, as follows:(19)$$\begin{array}{cc} \; & h_{i}\left(x\left(k+1\right)\right)=h_{i,0}+\left(h_{i,1,0}x_{1}\left(k+1\right)+h_{i,0,1}x_{2}\left(k+1\right)\right)\\ \; & +(h_{i,2,0}x_{1}^{2}\left(k+1\right)+h_{i,1,1}x_{1}\left(k+1\right)x_{2}\left(k+1\right)+h_{i,0,2}x_{2}^{2}\left(k+1\right))\\ \; & +(h_{i,3,0}x_{1}^{3}\left(k+1\right)+h_{i,2,1}x_{1}^{2}\left(k+1\right)x_{2}\left(k+1\right)+h_{i,1,2}x_{1}\left(k+1\right)x_{2}^{2}\left(k+1\right)+h_{i,3,0}x_{2}^{3}\left(k+1\right))\\ \; & +\cdots +\sum_{\begin{array}{c} l_{1}+l_{2}=l\\ l_{1,}l_{2}<l \end{array}}h_{i,l_{1},l_{2}}x_{1}^{l_{1}}\left(k+1\right)x_{2}^{l_{2}}\left(k+1\right)\\ \; & +\cdots +\sum_{\begin{array}{c} r_{1}+r_{2}=r\\ r_{1,}r_{2}<r \end{array}}h_{i,r_{1},r_{2}}x_{1}^{r_{1}}\left(k+1\right)x_{2}^{r_{2}}\left(k+1\right)\\ \; & +\gamma \left(x_{1}\left(k+1\right)\right)+\gamma \left(x_{2}\left(k+1\right)\right) \end{array}$$
where γ(x(k+1)) denotes the remainder variables of the measurement function: (20)$$\begin{array}{cc} \; & \left[\begin{array}{c} y_{1}^{\left(1\right)}\left(k+1\right)\\ y_{2}^{\left(2\right)}\left(k+1\right) \end{array}\right]\\ = & \left[\begin{array}{cccccc} h_{1}^{\left(1\right)} & h_{1}^{\left(2\right)} & \cdots & h_{1}^{\left(l\right)} & \cdots & \begin{array}{ccc} h_{1}^{\left(r\right)} & h_{1}\left(\gamma \left(x_{1}\right)\right) & h_{1}\left(\gamma \left(x_{2}\right)\right) \end{array}\\ h_{2}^{\left(1\right)} & h_{2}^{\left(2\right)} & \cdots & h_{2}^{\left(l\right)} & \cdots & \begin{array}{ccc} h_{2}^{\left(r\right)} & h_{2}\left(\gamma \left(x_{1}\right)\right) & h_{2}\left(\gamma \left(x_{2}\right)\right) \end{array} \end{array}\right]\left[\begin{array}{c} x^{\left(1\right)}\left(k+1\right)\\ x^{\left(2\right)}\left(k+1\right)\\ \vdots \\ x^{\left(l\right)}\left(k+1\right)\\ \vdots \\ x^{\left(r\right)}\left(k+1\right)\\ \gamma (x_{1}\left(k+1\right))\\ \gamma (x_{2}\left(k+1\right)) \end{array}\right]+\left[\begin{array}{c} v_{1}^{\left(1\right)}\left(k+1\right)\\ v_{2}^{\left(1\right)}\left(k+1\right) \end{array}\right]\\ = & \left[\begin{array}{cccccc} H^{\left(1\right)} & H^{\left(2\right)} & \cdots & H^{\left(l\right)} & \cdots & \begin{array}{cc} H^{\left(r\right)} & H(\gamma ) \end{array} \end{array}\right]\left[\begin{array}{c} x^{\left(1\right)}\left(k+1\right)\\ x^{\left(2\right)}\left(k+1\right)\\ \vdots \\ x^{\left(l\right)}\left(k+1\right)\\ \vdots \\ x^{\left(r\right)}\left(k+1\right)\\ \gamma (x(k+1)) \end{array}\right]+v^{\left(1\right)}\left(k+1\right)\\ = & H^{\left(1\right)}x^{\left(1\right)}\left(k+1\right)+\sum_{l=2}^{r}H^{\left(l\right)}x^{\left(l\right)}\left(k+1\right)+H\left(\gamma \right)\gamma \left(x\left(k+1\right)\right)+v^{\left(1\right)}\left(k+1\right) \end{array}$$

#### 3.2.3. Dynamic Modeling and Higher-Order Linearization of Remainder Variables Based on Taylor Expansion

In order to convert the pseudo-linearized model established above into a real high-order linearized form, it is necessary to expand the dimension and model using the high-order latent variables and remainder variables as new variables of the system.

In order to solve the problem in Remark 1, the high-order latent variable $x^{\left(l\right)}\left(k\right)$ and the remainder variables β(x(k)) and γ(x(k+1)) are used as time-varying parameters and the linear coupling relationship between the *l*-order latent variable $x^{\left(l\right)}\left(k+1\right)$ and the *u*-order latent variable $x^{\left(u\right)}\left(k\right)$ is established, along with the dynamic model relationship of the remainder variables β(x(k)) and γ(x(k+1)):(21)$$x^{\left(l\right)}\left(k+1\right)=A_{l}^{\left(u\right)}\left(k\right)x^{\left(u\right)}\left(k\right)\left(l,u=2,3,\cdots ,r\right)$$
(22)β(x(k+1))=A(β)β(x(k))
(23)γ(x(k+1))=A(γ)γ(x(k))

Then, the latent variables and remainder variables are used as the extension of the original state vector to realize the linear representation of the state model. Here, $A_{l}^{\left(u\right)}\left(k\right)$ and β(x(k+1)) can be identified according to the input information of the original state model; when there is no prior information, the settings are as follows:(24)$$A_{l}^{\left(u\right)}\left(k\right)=\left\{\begin{array}{cc} I, & l=u\\ 0, & l\neq u \end{array}\right.A\left(\beta \right)=I,A\left(\gamma \right)=I$$

Then,
(25)$$\left[\begin{array}{c} x^{\left(1\right)}\left(k+1\right)\\ x^{\left(2\right)}\left(k+1\right)\\ \vdots \\ x^{\left(l\right)}\left(k+1\right)\\ \vdots \\ \begin{array}{c} \begin{array}{c} x^{\left(r\right)}\left(k+1\right)\\ \beta (x(k+1)) \end{array}\\ \gamma (x(k+1)) \end{array} \end{array}\right]=\left[\begin{array}{ccc} A_{u}\left(k+1,k\right) & A(\beta ) & 0\\ 0 & A(\beta ) & 0\\ 0 & 0 & A(\gamma ) \end{array}\right]\left[\begin{array}{c} x^{\left(1\right)}\left(k\right)\\ x^{\left(2\right)}\left(k\right)\\ \vdots \\ x^{\left(l\right)}\left(k+1\right)\\ \vdots \\ \begin{array}{c} \begin{array}{c} x^{\left(r\right)}\left(k\right)\\ \beta (x(k)) \end{array}\\ \gamma (x(k)) \end{array} \end{array}\right]+\left[\begin{array}{c} w^{\left(1\right)}\left(k\right)\\ w^{\left(2\right)}\left(k\right)\\ \vdots \\ w^{\left(l\right)}\left(k\right)\\ \vdots \\ \begin{array}{c} w^{\left(r\right)}\left(k\right)\\ \begin{array}{c} w_{\beta }\left(k\right)\\ w_{\gamma }\left(k\right) \end{array} \end{array} \end{array}\right]$$

Denote $X\left(k\right)={\left[\begin{array}{cccc} {\left(x^{\left(1\right)}\left(k\right)\right)}^{T} & {\left(x^{\left(2\right)}\left(k\right)\right)}^{T} & \cdots & {\left(x^{\left(r\right)}\left(k\right)\right)}^{T}\begin{array}{cc} \begin{array}{cc} & {\left(\beta (x(k))\right)}^{T} \end{array} & {\left(\gamma (x(k))\right)}^{T} \end{array} \end{array}\right]}^{T}$
(26)$$\begin{array}{cc} \; & A\left(k+1,k\right)=\left[\begin{array}{ccc} A_{u}\left(k+1,k\right) & A(\beta ) & 0\\ 0 & A(\beta ) & 0\\ 0 & 0 & A(\gamma ) \end{array}\right], \end{array}$$
(27)$$\begin{array}{cc} \; & A_{u}\left(k+1,k\right)=\left[\begin{array}{cccccc} A_{1}^{\left(1\right)}\left(k+1,k\right) & A_{1}^{\left(2\right)}\left(k+1,k\right) & \cdots & A_{1}^{\left(u\right)}\left(k+1,k\right) & \cdots & A_{1}^{\left(r\right)}\left(k+1,k\right)\\ A_{2}^{\left(1\right)}\left(k+1,k\right) & A_{2}^{\left(2\right)}\left(k+1,k\right) & \cdots & A_{2}^{\left(u\right)}\left(k+1,k\right) & \cdots & A_{2}^{\left(r\right)}\left(k+1,k\right)\\ \vdots & \vdots & \ddots & \vdots & \ddots & \vdots \\ A_{l}^{\left(1\right)}\left(k+1,k\right) & A_{l}^{\left(2\right)}\left(k+1,k\right) & \cdots & A_{l}^{\left(u\right)}\left(k+1,k\right) & \cdots & A_{l}^{\left(r\right)}\left(k+1,k\right)\\ \vdots & \vdots & \ddots & \vdots & \ddots & \vdots \\ A_{r}^{\left(1\right)}\left(k+1,k\right) & A_{r}^{\left(2\right)}\left(k+1,k\right) & \cdots & A_{r}^{\left(u\right)}\left(k+1,k\right) & \cdots & A_{r}^{\left(r\right)}\left(k+1,k\right) \end{array}\right] \end{array}$$
(28)$$\begin{array}{cc} \; & W\left(k\right)={\left[\begin{array}{cccccccc} {\left(w^{\left(1\right)}\left(k\right)\right)}^{T} & {\left(w^{\left(2\right)}\left(k\right)\right)}^{T} & \cdots & {\left(w^{\left(l\right)}\left(k\right)\right)}^{T} & \cdots & {\left(w^{\left(r\right)}\left(k\right)\right)}^{T} & {\left(w_{\beta }\left(k\right)\right)}^{T} & {\left(w_{\gamma }\left(k\right)\right)}^{T} \end{array}\right]}^{T} \end{array}$$

The argmented state equation can be described as
(29)X(k+1)=A(k+1,k)X(k)+W(k)

Based on the above-mentioned argmented state variables, the linearized description model of the observation equation is similarly described as follows:(30)$$\begin{array}{cc} \; & \left[\begin{array}{c} y_{1}^{\left(1\right)}\left(k+1\right)\\ y_{2}^{\left(2\right)}\left(k+1\right) \end{array}\right]\\ = & \left[\begin{array}{cccccc} h_{1}^{\left(1\right)} & h_{1}^{\left(2\right)} & \cdots & h_{1}^{\left(l\right)} & \cdots & \begin{array}{ccc} h_{1}^{\left(r\right)} & h_{1}\left(\beta \left(x\right)\right) & h_{1}\left(\gamma \left(x\right)\right) \end{array}\\ h_{2}^{\left(1\right)} & h_{2}^{\left(2\right)} & \cdots & h_{2}^{\left(l\right)} & \cdots & \begin{array}{ccc} h_{2}^{\left(r\right)} & h_{2}\left(\beta \left(x\right)\right)) & h_{2}\left(\gamma \left(x\right)\right) \end{array} \end{array}\right]\left[\begin{array}{c} x^{\left(1\right)}\left(k+1\right)\\ x^{\left(2\right)}\left(k+1\right)\\ \vdots \\ x^{\left(l\right)}\left(k+1\right)\\ \vdots \\ x^{\left(r\right)}\left(k+1\right)\\ \beta (x(k+1))\\ \gamma (x(k+1)) \end{array}\right]+\left[\begin{array}{c} v_{1}^{\left(1\right)}\left(k+1\right)\\ v_{2}^{\left(1\right)}\left(k+1\right) \end{array}\right]\\ \; & \;=\left[\begin{array}{cccccc} H^{\left(1\right)} & H^{\left(2\right)} & \cdots & H^{\left(l\right)} & \cdots & \begin{array}{cc} \begin{array}{cc} H^{\left(r\right)} & H(\beta ) \end{array} & H(\gamma ) \end{array} \end{array}\right]\left[\begin{array}{c} x^{\left(1\right)}\left(k+1\right)\\ x^{\left(2\right)}\left(k+1\right)\\ \vdots \\ x^{\left(l\right)}\left(k+1\right)\\ \vdots \\ x^{\left(r\right)}\left(k+1\right)\\ \beta (x(k+1))\\ \gamma (x(k+1)) \end{array}\right]+v^{\left(1\right)}\left(k+1\right) \end{array}$$

Denote $H\left(k+1\right)=\left[\begin{array}{ccc} H_{u}\left(k+1\right) & H(\beta ) & H(\gamma ) \end{array}\right]$,

$H_{u}\left(k+1\right)=\left[\begin{array}{cccccc} H^{\left(1\right)} & H^{\left(2\right)} & \cdots & H^{\left(u\right)} & \cdots & H^{\left(r\right)} \end{array}\right]$,$V\left(k+1\right)=\left[\begin{array}{c} v_{1}\left(k+1\right)\\ v_{2}\left(k+1\right) \end{array}\right]$ where, H(β)=0,H(γ)=I.

Then, the linearized representation of the nonlinear measurement model equation is as follows:(31)Y(k+1)=H(k+1)X(k+1)+V(k+1)

### 3.3. Recursive Filtering Algorithm

Considering the linearized state model and measurement model Equations (30) and (32) shown above, its statistical characteristics are as follows:(32)$$E\left\{W\left(k\right)W^{T}\left(k\right)\right\}=Q_{w}\left(k\right)$$
(33)$$E\left\{V\left(k+1\right)V^{T}\left(k+1\right)\right\}=R_{V}\left(k+1\right)$$
(34)$$E\{W\left(k\right)V^{T}\left(k+1\right)\}=0$$

**Step 1**: Set the initial values of the new system; then, according to the initial value x(k) of x(0), the following formula is satisfied:(35)$$E\left\{x\left(0\right)\right\}={\hat{x}}_{0}$$
(36)$$E\left\{\left[x\left(0\right)-x_{0}\right]{\left[x\left(0\right)-x_{0}\right]}^{T}\right\}=P_{0}$$

Then, $X_{0}$ satisfies the following characteristics:(37)$$X\left(0\right)={\left[\begin{array}{cccc} {\left(x^{\left(1\right)}\left(0\right)\right)}^{T} & {\left(x^{\left(2\right)}\left(0\right)\right)}^{T} & \cdots & {\left(x^{\left(r\right)}\left(0\right)\right)}^{T}\begin{array}{cc} \begin{array}{cc} & {\left(\beta (x(0))\right)}^{T} \end{array} & {\left(\gamma (x(0))\right)}^{T} \end{array} \end{array}\right]}^{T}$$
(38)$${\bar{P}}_{0}=diag\left\{\left[\begin{array}{cccccc} P_{0}^{\left(1\right)} & P_{0}^{\left(2\right)} & \cdots & P_{0}^{\left(r\right)} & P_{0}\left(\beta_{0}\right) & P_{0}\left(\gamma_{0}\right) \end{array}\right]\right\}$$
(39)$$E\left\{\left[X\left(0\right)-{\hat{X}}_{0}\right]{\left[X\left(0\right)-{\hat{X}}_{0}\right]}^{T}\right\}={\bar{P}}_{0}$$
where ${\bar{P}}_{0}=diag\left\{\left[\begin{array}{cccccc} P_{0}^{\left(1\right)} & P_{0}^{\left(2\right)} & \cdots & P_{0}^{\left(r\right)} & P_{0}\left(\beta_{0}\right) & P_{0}\left(\gamma_{0}\right) \end{array}\right]\right\}\geq 0$ is a positive semi-definite matrix.

**Step 2**: Recursive filtering

Assuming that y(1),y(2),⋯,y(k) has been obtained, that is, $\hat{X}\left(k|k\right)$ and $\bar{P}\left(k|k\right)$ are known, the new higher-order Kalman filter is designed as follows: (40)$$\begin{array}{cc} \hat{X}\left(k+1|k+1\right) & =E\{X\left(k+1\right)|{\hat{X}}_{0},y\left(1\right),y\left(2\right),\cdots ,y\left(k\right),y\left(k+1\right)\}\\ \; & =E\{X\left(k+1\right)|\hat{X}\left(k|k\right),y\left(k+1\right)\} \end{array}$$

The corresponding covariance matrix is as follows:(41)$$\bar{P}\left(k+1|k+1\right)=E\left\{\left[X\left(k+1\right)-\hat{X}\left(k+1|k+1\right)\right]{\left[X\left(k+1\right)-\hat{X}\left(k+1|k+1\right)\right]}^{T}\right\}$$

**Step 3**: Time Update

(1) Obtain the following based on $\hat{X}\left(k|k\right)$ and A(k+1,k):(42)$$\hat{X}\left(k+1|k\right)=A\left(k+1,k\right)\hat{X}\left(k|k\right)$$

(2) Obtain $\bar{P}\left(k+1|k\right)$ based on $\bar{P}\left(k|k\right)$ and $Q_{W}\left(k\right)$:(43)$$\bar{P}\left(k+1|k\right)=A\left(k+1|k\right)\bar{P}\left(k|k\right)A^{T}\left(k+1|k\right)+Q_{W}\left(k\right)$$

**Step 4**: Observe Update

(3) According to the relevant information of $\bar{P}\left(k+1|k\right)$ and the observed value, the gain matrix K(k+1) is obtained:(44)$$\hat{Y}\left(k+1|k\right)=H\left(k+1\right)\hat{X}\left(k+1|k\right)$$
(45)$$K\left(k+1\right)=\bar{P}\left(k+1|k\right)H^{T}\left(k+1\right)*{\left[H\left(k+1\right)\bar{P}\left(k+1|k\right)H^{T}\left(k+1\right)+R_{V}\left(k+1\right)\right]}^{-1}$$

(4) A higher-order Kalman filter is obtained from the remainders of the actual and predicted observations of $\hat{X}\left(k+1|k\right)$ and K(k+1):(46)$$\hat{X}\left(k+1|k+1\right)=\hat{X}\left(k+1|k\right)+K\left(k+1\right)\left[Y\left(k+1\right)-H\left(k+1\right)\hat{X}\left(k+1|k\right)\right]$$

(5) Calculate the update error covariance matrix:(47)$$\bar{P}\left(k+1|k+1\right)=\left(I-K\left(k+1\right)H\left(k+1\right)\right)\bar{P}\left(k+1|k\right)$$

$\hat{X}\left(k+1|k+1\right)$ is the obtained state estimate.

## 4. Performance Analysis

### 4.1. Projection Matrix Analysis

Rewrite the expanded state variables in Equation (Equation 46) as the original state variables and all higher-order variables and remainder variables in the following form: (48)$$\begin{array}{cc} \left[\begin{array}{c} \hat{x}\left(k+1|k+1\right)\\ \beta (\hat{x}\left(k+1|k+1\right)) \end{array}\right]\; & =\left[\begin{array}{c} \hat{x}\left(k+1|k\right)\\ \beta (\hat{x}\left(k+1|k\right)) \end{array}\right]\\ \; & +\left[\begin{array}{c} K_{x}\left(k+1\right)\\ K_{\beta }\left(k+1\right) \end{array}\right]*\left(\left[\begin{array}{cc} h_{x}\left(k+1\right) & \gamma (h_{x}\left(k+1\right)) \end{array}\right]\left[\begin{array}{c} e(k+1|k)\\ \beta (\hat{x}\left(k+1|k\right)) \end{array}\right]+v\left(k+1\right)\right) \end{array}$$
where $K_{x}\left(k+1\right)$ and $K_{\beta }\left(k+1\right)$ are the gain matrices corresponding to the original variables and the remainder augmented-dimensional variables, respectively, and $h_{x}\left(k+1\right)$ and $\gamma (h_{x}\left(k+1\right))$ are the observation matrices corresponding to the original variables and the remaining augmented-dimensional variables, respectively. Let the projection operator be
(49)$$P_{e}=\left[\begin{array}{cc} I_{x} & O_{\beta } \end{array}\right]$$
where $I_{x}$ and $O_{\beta }$ are matrices that match the dimensions of the original and remainder augmented variables, respectively.

Then,
(50)$$\hat{x}\left(k+1|k+1\right)=P_{e}*\hat{X}\left(k+1|k+1\right)=\left[\begin{array}{cc} I_{x} & O_{\beta } \end{array}\right]*\left[\begin{array}{c} \hat{x}\left(k+1|k+1\right)\\ \beta (\hat{x}\left(k+1|k+1\right)) \end{array}\right]$$

Substituting Equation (Equation 48) into Equation (Equation 50), we obtain
(51)$$\begin{array}{cc} \hat{x}\left(k+1|k+1\right)\; & =\hat{x}\left(k+1|k\right)+K_{x}\left(k+1\right)*[h_{x}\left(k+1\right)e\left(k+1|k\right)\\ \; & +\gamma \left(h_{x}\left(k+1\right)\right)\beta \left(\hat{x}\left(k+1|k\right)\right)+v\left(k+1\right)]\\ \; & =\hat{x}\left(k+1|k\right)+K_{x}\left(k+1\right)*h_{x}\left(k+1\right)e\left(k+1|k\right)\\ \; & +K_{x}\left(k+1\right)*\gamma \left(h_{x}\left(k+1\right)\right)\beta \left(\hat{x}\left(k+1|k\right)\right)+K_{x}\left(k+1\right)v\left(k+1\right) \end{array}$$

It is found that after projection through the projection matrix, only the original system state estimation value is retained, which reduces the complexity of the algorithm and the actual computational burden, and includes more information from the model, improving the estimation accuracy.

### 4.2. Covariance Matrix Analysis

Bring Equation (Equation 48) into Equation (Equation 47) and at the same time divide the covariance matrix into blocks according to the original variables and the remaining dimension expansion variables, then write as follows:(52)$$\begin{array}{cc} \; & \left[\begin{array}{cc} P_{x}\left(k+1|k+1\right) & P_{x\beta }\left(k+1|k+1\right)\\ P_{\beta x}\left(k+1|k+1\right) & P_{\beta }\left(k+1|k+1\right) \end{array}\right]\\ \; & \;=\left(\left[\begin{array}{cc} I_{x} & O\\ O & I_{\beta } \end{array}\right]-\left[\begin{array}{c} K_{x}\left(k+1\right)\\ K_{\beta }\left(k+1\right) \end{array}\right]\left[\begin{array}{cc} h_{x}\left(k+1\right) & \gamma (h_{x}\left(k+1\right)) \end{array}\right]\right)*\left[\begin{array}{cc} P_{x}\left(k+1|k\right) & P_{x\beta }\left(k+1|k\right)\\ P_{\beta x}\left(k+1|k\right) & P_{\beta }\left(k+1|k\right) \end{array}\right] \end{array}$$
where $P_{x}\left(k+1|k+1\right)$ is the covariance matrix of the original variable,

$P_{x\beta }\left(k+1|k+1\right)$ is the covariance matrix of the original variable and the remainder variable, $P_{\beta x}\left(k+1|k+1\right)$ is the covariance matrix of the remainder variable and the original variable, and $P_{\beta }\left(k+1|k+1\right)$ is the covariance matrix of the remainder variable.

Then, the original variable covariance matrix can be calculated by the projection matrix as follows: (53)$$\begin{array}{cc} \; & P_{x}\left(k+1|k+1\right)\\ \; & =P_{e}*\bar{P}\left(k+1|k+1\right)*P_{e}^{T}\\ \; & =\left[\begin{array}{cc} I_{x} & O_{\beta } \end{array}\right]*\left[\begin{array}{cc} P_{x}\left(k+1|k+1\right) & P_{x\beta }\left(k+1|k+1\right)\\ P_{\beta x}\left(k+1|k+1\right) & P_{\beta }\left(k+1|k+1\right) \end{array}\right]*\left[\begin{array}{c} I_{x}\\ O_{\beta } \end{array}\right]\\ \; & =\left[\begin{array}{cc} I_{x} & O_{\beta } \end{array}\right]*\left(I-\left[\begin{array}{c} K_{x}\left(k+1\right)\\ K_{\beta }\left(k+1\right) \end{array}\right]*\left[\begin{array}{cc} h_{x}\left(k+1\right) & \gamma (h_{x}\left(k+1\right)) \end{array}\right]\right)\\ \; & *\left[\begin{array}{cc} P_{x}\left(k+1|k\right) & P_{x\beta }\left(k+1|k\right)\\ P_{\beta x}\left(k+1|k\right) & P_{\beta }\left(k+1|k\right) \end{array}\right]*\left[\begin{array}{c} I_{x}\\ O_{\beta } \end{array}\right]\\ \; & =\left(I_{x}-K_{x}\left(k+1\right)*h_{x}\left(k+1\right)\right)*P_{x}\left(k+1|k\right)-K_{x}\left(k+1\right)*\gamma \left(h_{x}\left(k+1\right)\right)*P_{x\beta }\left(k+1|k\right) \end{array}$$

In the experiment, it was found that $K_{x}\left(k+1\right)*\gamma \left(h_{x}\left(k+1\right)\right)*P_{x\beta }\left(k+1|k\right)$ is a positive-definite matrix. Compared with the original EKF algorithm, the posterior covariance matrix becomes smaller with the application of higher-order residual information, and the algorithm makes full use of the higher-order information of the model. Theoretically, the state estimation accuracy will be higher.

## 5. Simulation Experiments

For multi-robot formation systems, the state equation is as follows [24,35]:(54)$$\xi_{i}\left(k+1\right)=\xi_{i}\left(k\right)+\phi_{i}\left(k\right)\cos\theta_{i}\left(k\right)+c\sum_{j=1}^{N}\Gamma \xi_{j}\left(k\right)+\omega_{i}^{x}\left(k\right)$$
(55)$$\psi_{i}\left(k+1\right)=\psi_{i}\left(k\right)+\phi_{i}\left(k\right)\sin\theta_{i}\left(k\right)+c\sum_{j=1}^{N}\Gamma \psi_{j}\left(k\right)+\omega_{i}^{y}\left(k\right)$$
(56)$$\theta_{i}\left(k+1\right)=\theta_{i}\left(k\right)+\delta_{i}\left(k\right)+c\sum_{j=1}^{N}\Gamma \theta_{j}\left(k\right)+\omega_{i}^{\theta }\left(k\right)$$
where $\left(\xi_{i}\left(k\right),\psi_{i}\left(k\right)\right)$ and $\theta_{i}\left(k\right)$ represent the position and direction of the *i*-th robot, respectively, and $\phi_{i}\left(k\right)$ and $\delta_{i}\left(k\right)$ represent the linear velocity and angular velocity, respectively. Suppose that $\phi_{i}\left(k\right)=0.15$, $\delta_{i}\left(k\right)=0.3$, $\omega_{i}\left(k\right)=\left(\omega_{i}^{x}\left(k\right),\omega_{i}^{y}\left(k\right),\omega_{i}^{\theta }\left(k\right)\right)$ are Gaussian white noise with zero mean variance $Q_{i}\left(k\right)$.

In a multi-robot formation, Γ is a time-varying matrix related to the formation and Γ is the formation; if Γ is an upper triangular matrix, it means that the formation has a fixed movement formation, as shown in Figure 2. If Γ=Γ(k), it means that the formation is time-varying and has a random switching topology, as shown in Figure 3. We use the extended Kalman filter (EKF) and high-order Kalman filter methods to estimate the second-order Kalman filter algorithm (SOKF), including the remainder extended Kalman filter algorithm (REKF), as well as the second-order remainder Kalman filter algorithm(SEORKF) for comparison.

As can be seen from the above figure, in a robot formation with a fixed formation, the robot maintains the formation at the first moment and continues to move with a unique nonlinear model. However, in a formation with a random switching topology, at the first moment it maintains a fixed formation structure. With the constraints of different communication or task division, the formation at each moment will be different. There are arbitrary formations or individual formations, and they maintain their own nonlinear motion.

The visual measurement is provided by the following equation:(57)$$p_{i}\left(k\right)=\frac{\gamma_{u}}{z_{f}^{c}}\left[-\left(s_{x}-\xi_{i}\left(k\right)\right)\sin\theta_{i}\left(k\right)+\left(s_{x}-\psi_{i}\left(k\right)\right)\cos\theta_{i}\left(k\right)-d_{2}\right]+p_{0}+v_{i}^{p}\left(k\right)$$
(58)$$q_{i}\left(k\right)=\frac{\gamma_{v}}{z_{f}^{c}}\left[-\left(s_{y}-\xi_{i}\left(k\right)\right)\cos\theta_{i}\left(k\right)-\left(s_{y}-\psi_{i}\left(k\right)\right)\sin\theta_{i}\left(k\right)-d_{1}\right]+q_{0}+v_{i}^{q}\left(k\right)$$
where $\left(p_{i}\left(k\right),q_{i}\left(k\right)\right)$ defines the coordinates of the robot in the image plane, $\left(d_{1},d_{2}\right)$ are its frame coordinates, $z_{f}^{c}$ is the distance from the optical center of the camera to the robot, and $\gamma_{u}$ and $\gamma_{v}$ are variable pixel magnification factors; for a visual tracking system, the feature points are placed on the ceiling, $\left(s_{x},s_{y}\right)$ are the coordinates of the feature points in the world frame, and $\left(p_{0},q_{0}\right)$ is the camera image coordinate principal point; $v_{i}\left(k\right)=\left(v_{i}^{p}\left(k\right),v_{i}^{q}\left(k\right)\right)$ is white Gaussian noise with zero mean variance $R_{i}\left(k\right)$.

In the simulation, the following parameters of the visual tracking system are adopted:

$d_{1}=-0.0668$,$d_{2}=0.0536$,$z_{f}^{c}=2.1050$,$\gamma_{u}=902.13283$,$\gamma_{v}=902.50141$,$p_{0}=347.20436$,

$q_{0}=284.34705.$ The process noise covariance matrix is $Q_{i}\left(k+1\right)=diag\{0.01,0.01,0.01\},$ and the measurement noise covariance matrix is $R_{i}\left(k\right)=diag\left\{25^{2},25^{2}\right\}\;(i=1,2,3)$.

To illustrate the tracking performance of the proposed filter, we use the root mean square error (MSE) of the three robots. Over 50 Monte-Carlo runs were obtained, and the table below shows the mean squared error for each robot position:(59)$$\begin{array}{cc} \; & MSE_{\xi }\left(k\right)=\frac{1}{3}\sum_{i=1}^{3}\frac{1}{50}\sum_{m=1}^{50}{\left(\xi_{i}\left(k\right)-{\hat{\xi }}_{i}^{\left(m\right)}\left(k\right)\right)}^{2}\\ \; & MSE_{\psi }\left(k\right)=\frac{1}{3}\sum_{i=1}^{3}\frac{1}{50}\sum_{m=1}^{50}{\left(\psi_{i}\left(k\right)-{\hat{\psi }}_{i}^{\left(m\right)}\left(k\right)\right)}^{2} \end{array}$$
where $\left({\hat{\xi }}_{i}^{\left(m\right)}\left(k\right),{\hat{\psi }}_{i}^{\left(m\right)}\left(k\right)\right)$ represents the position estimate of the *i*-th robot at the *m*-th Monte Carlo run.

### 5.1. Trajectory Estimation of Multi-Robot in Fixed Formation

In a multi-robot formation, Γ is a time-varying matrix related to the formation, Γ is the formation, and when Γ is an upper triangular matrix, it means that the formation has a fixed movement formation to complete a certain task. Table 1 shows the multi-robot trajectory estimation MSE with fixed formation, where ’Improved’ refers to the improved estimation accuracy compared with EKF method.

The original nonlinear filtering algorithms based on Taylor expansion, such as extended Kalman filtering and high-order Kalman filtering methods, directly discard the truncation error. The estimation accuracy has a great influence. Table 2 shows the truncation errors generated by the EKF and second-order Kalman filtering methods for estimating the fixed formation trajectories of multi-robot. It can be seen from the table below that both methods have considerable truncation errors.

In the high-order Kalman filtering method, the remainder variables are introduced into the algorithm to make full use of the nonlinear model information, which can effectively avoid the influence of the truncation error on the estimation results; however, there will always be rounding errors. Here, the rounding error results generated by the first-order remainder extended Kalman filtering algorithm and the second-order remainder Kalman filtering algorithm with a fixed formation of robots are compared and analyzed, as show in Table 3, with the result that the rounding error is much smaller than the truncation error.

As follows, in the fixed formation, Figure 4 shows the real trajectory of the fixed multi-robot formation; Figure 5a,b shows the R1 estimation error in X-Coordinate and Y-Coordinate respectively; Figure 6 shows the histogram of R1 estimation error; Figure 7a,b shows the R2 estimation error in X-Coordinate and Y-Coordinate respectively; Figure 8 shows the histogram of R2 estimation error; Figure 9a,b shows the R3 estimation error in X-Coordinate and Y-Coordinate respectively; Figure 10 shows the histogram of R3 estimation error.

### 5.2. Multi-Robot Formation Trajectory Estimation with Random Topology Structure

In the multi-robot formation, Γ is the time-varying matrix related to the formation, that is, Γ=Γ(k), indicating that the formation has a random switching topology. Table 4 shows MSE for multi-robot trajectory estimation and Table 5 shows the truncation error of trajectory estimation. The truncation error of the trajectory estimation extended Kalman algorithm and the second-order Kalman filtering algorithm in Table 6 shows the rounding error when the trajectory estimation is performed by introducing the remainder variables into the above two algorithms, and the results are similar to the trajectory estimation under the fixed formation.

As follows, in random topology formation, Figure 11 shows the real trajectory; Figure 12a,b shows the R1 estimation error in X-Coordinate and Y-Coordinate respectively; Figure 13 shows the histogram of R1 estimation error; Figure 14a,b shows the R2 estimation error in X-Coordinate and Y-Coordinate respectively; Figure 15 shows the histogram of R2 estimation error; Figure 16a,b shows the R3 estimation error in X-Coordinate and Y-Coordinate respectively; Figure 17 shows the histogram of R3 estimation error.

As shown in the figure above, we use a fixed formation formation and a random topology formation to respectively estimate the actual trajectories of the three robots, use the estimation error of each robot in the x–y direction of the two-dimensional plane as a measure, and use the EKF and the second-order Kalman filtering(SOKF) methods, respectively, to introduce the residual term variable in each of these two methods. From the results, it can be concluded that the estimation accuracy of EKF is the lowest, followed by the second-order Kalman filtering algorithm (SOKF), the remainder extended Kalman filtering Algorithm (REKF), and the remainder second-order Kalman filtering algorithm (SORKF). The other three filtering methods designed according to EKF structure, excepting EKF, significantly improve the estimation accuracy of EKF. Thus, we can see that for the robot in the motion model, the degree of non-linearity of the turning characteristics is relatively high. If first-order Taylor expansion is performed, the information contained in it cannot be completely extracted, which inevitably leads to a decrease in the estimation accuracy. The higher the degree of expansion performed, the higher the robot position estimation accuracy, and the more accurate the robot estimation estimation. In addition, we analyzed the truncation error of the EKF and second-order Kalman filtering(SOKF) algorithms. The truncation error has a great influence on the estimation accuracy. When the nonlinearity is strong, the model information cannot be fully utilized by directly discarding the truncation error, resulting in low estimation accuracy. After the variables are introduced, the rounding error of the estimation results is analyzed, and it is found that the rounding error is much smaller than the truncation error. The Taylor expansion of the nonlinear model and the use of remainder variables can make full use of the model information to improve the estimation accuracy and make the estimation more accurate.

## 6. Conclusions and Future Work

This paper studies the multi-robot trajectory estimation problem, and proposes a fusion estimation method based on the high-order Kalman filter algorithm. The extended Kalman filter (EKF) algorithm used in the existing robot position estimation only considers first-order expansion and ignores the high-order information. To solve the problem, a joint trajectory estimation method with a multi-robot formation based on the high-order Kalman filtering method was adopted, the Taylor expansion of the state equation and the observation equation was carried out, and the remainder variables were introduced on this basis, effectively improving the estimation accuracy. Through a simulation, we found that for robot fixed formations and random topology formations the introduction of remainder variables improves the accuracy of position estimation by more than 50%, and in certain cases even more, compared to the EKF algorithm. At the same time, the truncation error in the estimation process of the EKF and the high-order Kalman filter algorithms was analyzed and compared with the rounding error of the estimation algorithm after the introduction of the remainder variable. We found that the rounding error is much smaller than the truncation error. After the remainder variable is introduced during filtering, the algorithm makes full use of the model information, and the estimation accuracy is greatly improved.

In future research, we may encounter more complex models. If the state model has strong nonlinear characteristics, the observation model has super strong nonlinear characteristics, and the modeling error is non Gaussian white noise, our method will be difficult to achieve, and will be completed with the help of the characteristic function filtering method [36,37]; At the same time, for the neural network model, we can also introduce the high-order Kalman filtering method to update the real-time parameters [38]; Finally, for the state estimation of non-cooperative targets, there must be multiple targets and sensors. We can extend the method in this paper to distributed filtering. Similarly, for the distributed model of Federated learning, we will also applicable [39,40].

## Figures and Tables

**Figure 1 sensors-22-05590-f001:**
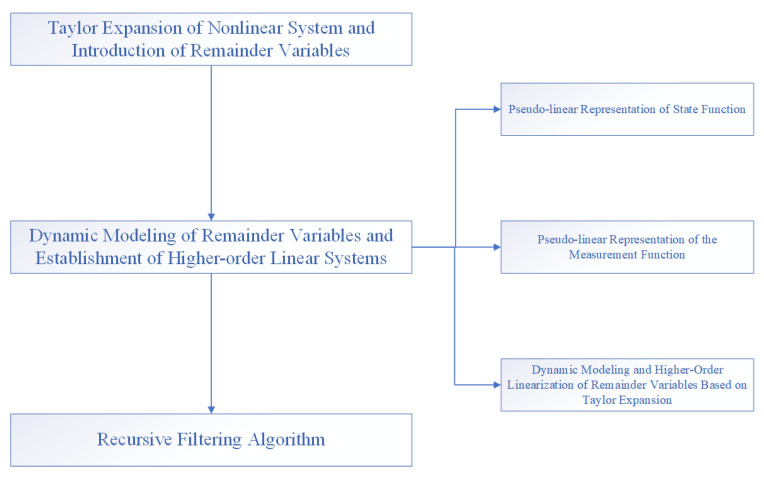
Algorithm block diagram.

**Figure 2 sensors-22-05590-f002:**
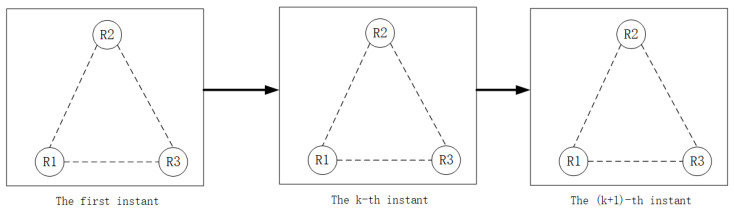
Multi-robot formation with fixed formation.

**Figure 3 sensors-22-05590-f003:**
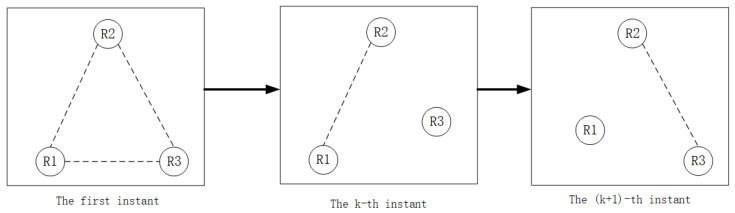
Multi-robot formation in a random topology formation.

**Figure 4 sensors-22-05590-f004:**
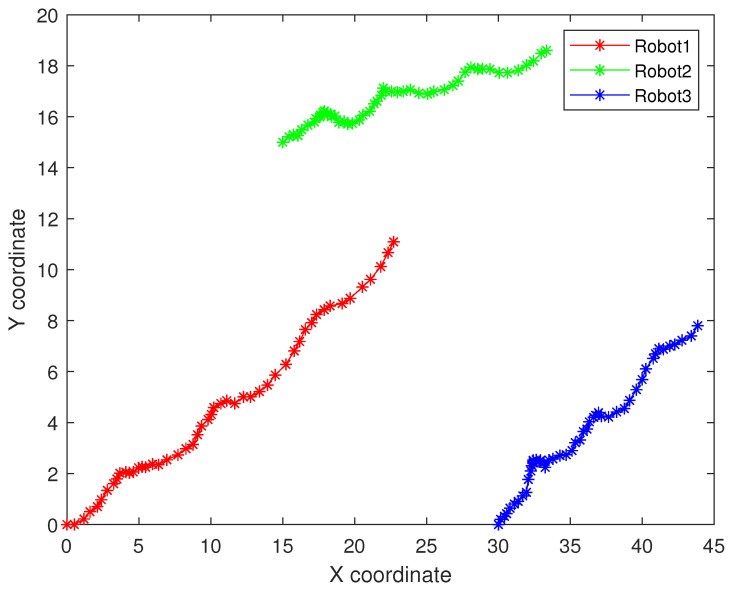
The real trajectory of the fixed multi-robot formation.

**Figure 5 sensors-22-05590-f005:**
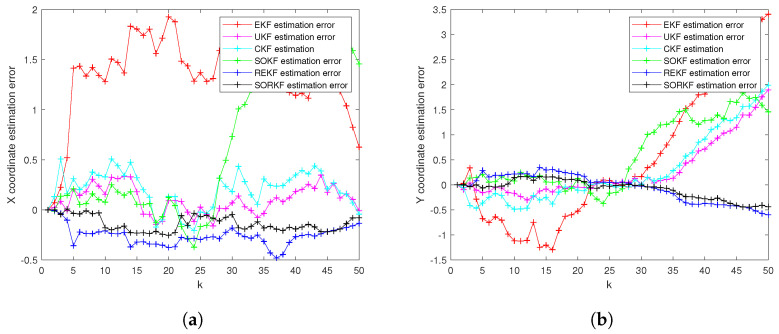
R1 estimation error of multi-robot in fixed formation. (**a**) X-Coordinate estimation error; (**b**) Y-Coordinate estimation error.

**Figure 6 sensors-22-05590-f006:**
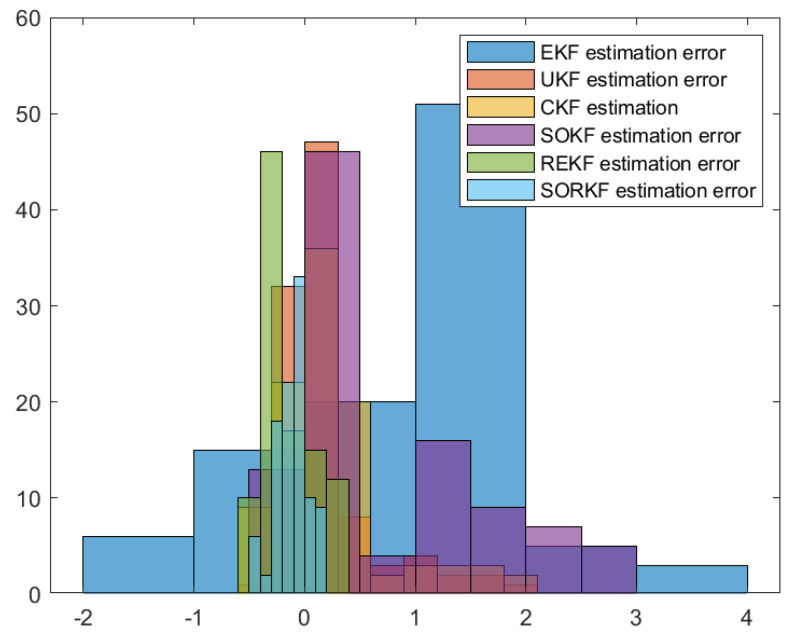
Histogram of R1 positioning estimation error in fixed formation.

**Figure 7 sensors-22-05590-f007:**
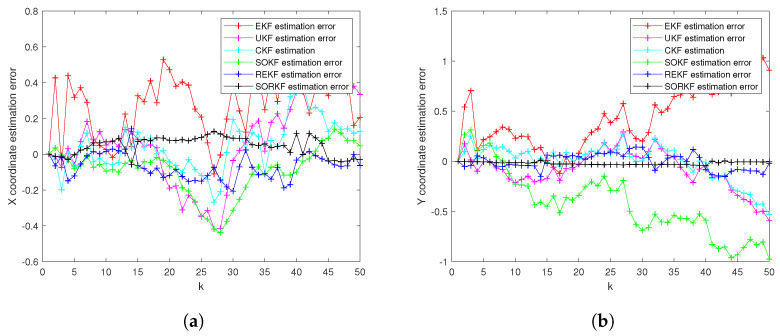
R2 estimation error of multi-robot in fixed formation. (**a**) X-Coordinate estimation error; (**b**) Y-Coordinate estimation error.

**Figure 8 sensors-22-05590-f008:**
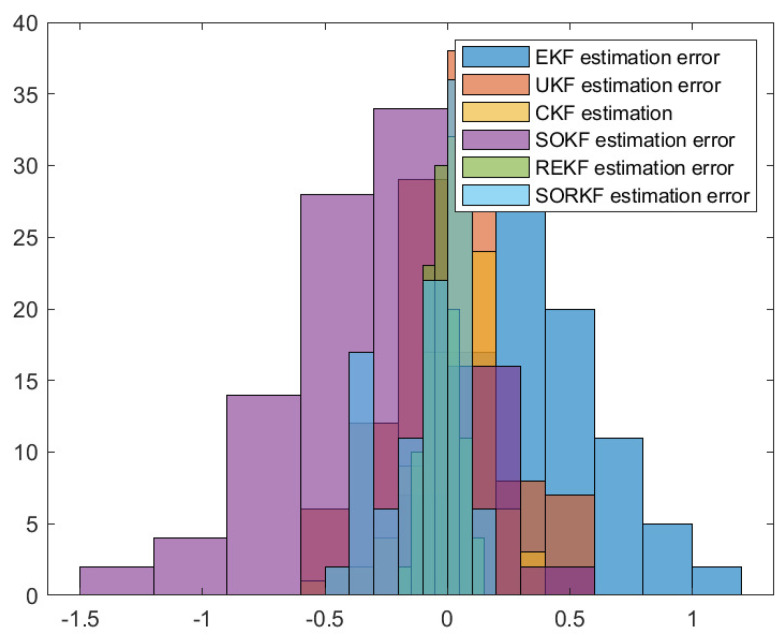
Histogram of R2 positioning estimation error in fixed formation.

**Figure 9 sensors-22-05590-f009:**
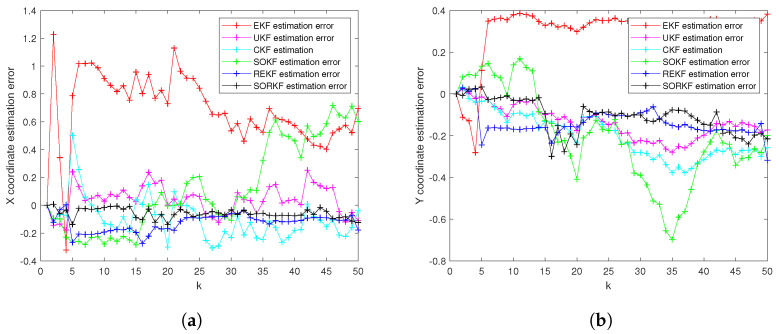
R3 estimation error of multi-robot in fixed formation. (**a**) X-Coordinate estimation error; (**b**) Y-Coordinate estimation error.

**Figure 10 sensors-22-05590-f010:**
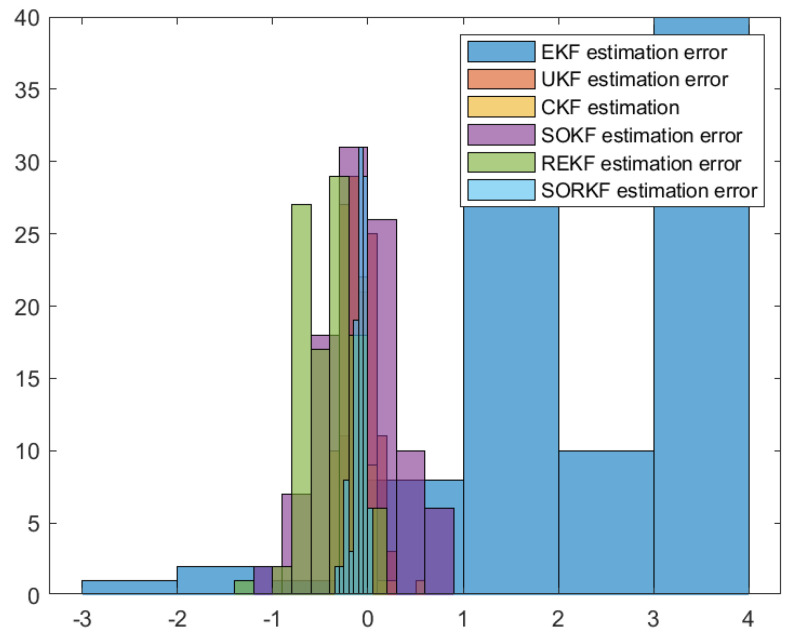
Histogram of R3 positioning estimation error in fixed formation.

**Figure 11 sensors-22-05590-f011:**
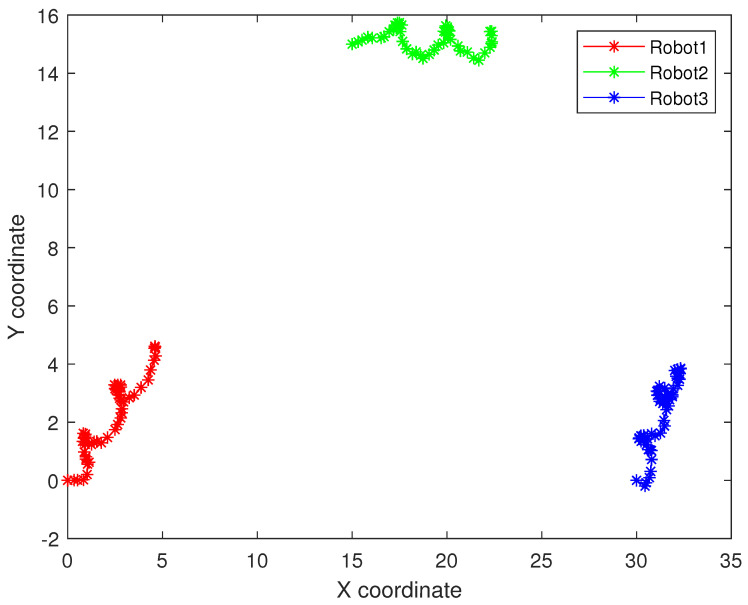
The real trajectory of the multi-robot swarm in random topology formation.

**Figure 12 sensors-22-05590-f012:**
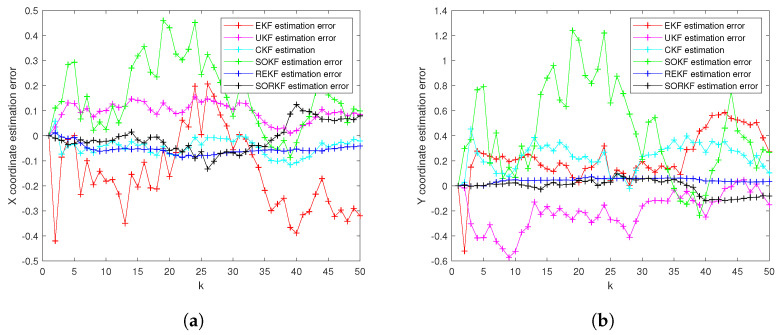
R1 estimation error of multi-robot swarm in random topology formation. (**a**) X-Coordinate estimation error; (**b**) Y-Coordinate estimation error.

**Figure 13 sensors-22-05590-f013:**
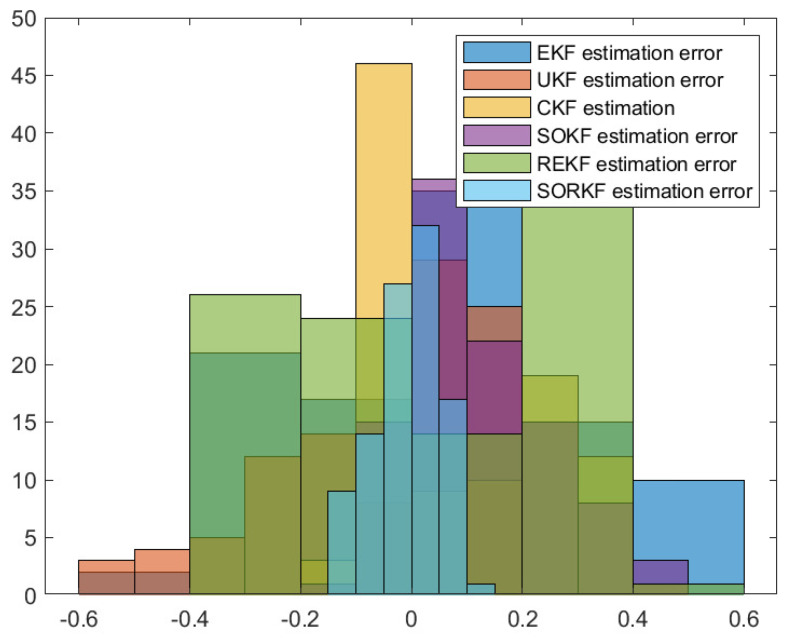
Histogram of R1 positioning estimation error in random topology formation.

**Figure 14 sensors-22-05590-f014:**
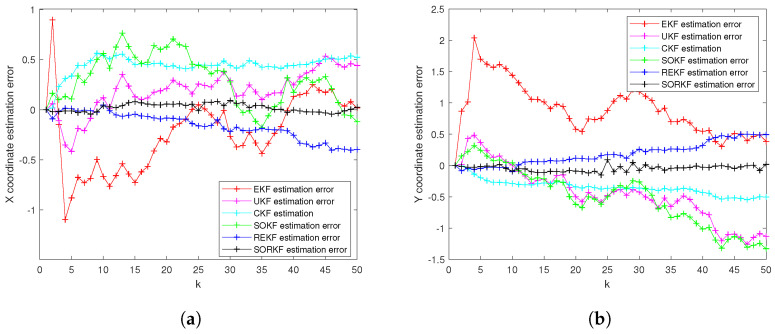
R2 estimation error of multi-robot swarm in random topology formation. (**a**) X-Coordinate estimation error; (**b**) Y-Coordinate estimation error.

**Figure 15 sensors-22-05590-f015:**
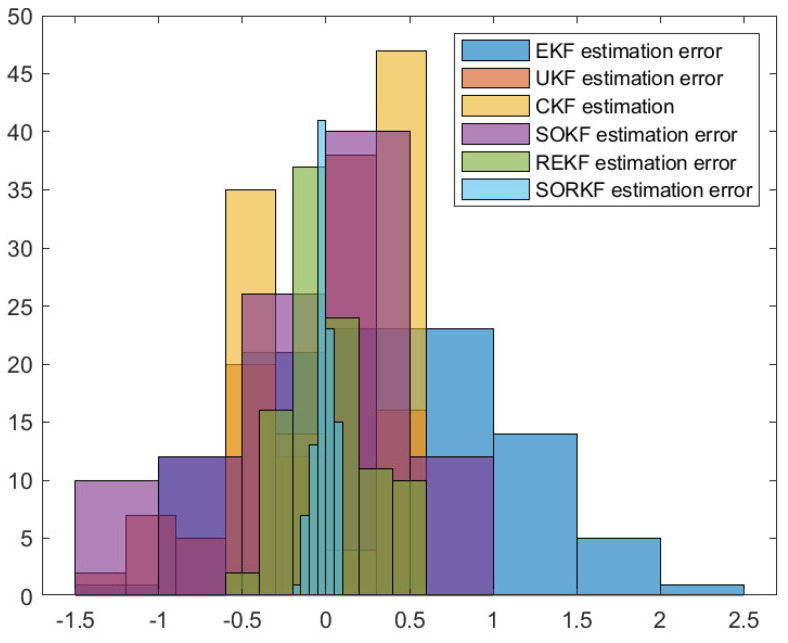
Histogram of R2 positioning estimation error in random topology formation.

**Figure 16 sensors-22-05590-f016:**
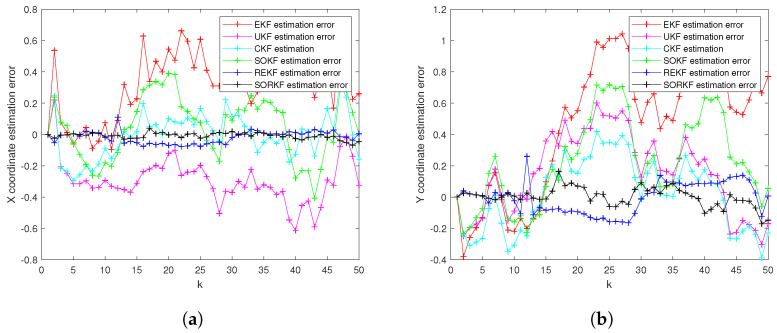
R3 estimation error of multi-robot swarm in random topology formation. (**a**) X-Coordinate estimation error; (**b**) Y-Coordinate estimation error.

**Figure 17 sensors-22-05590-f017:**
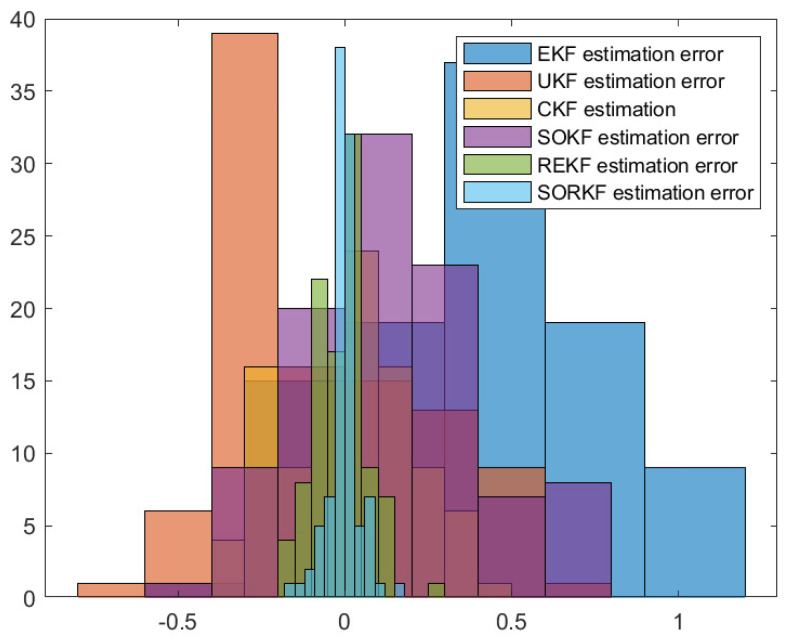
Histogram of R3 positioning estimation error in random topology formation.

**Table 1 sensors-22-05590-t001:** MSE for multi-robot trajectory estimation in fixed formation.

R	MSE of R1		MSE of R2		MSE of R3	
Method	x	y	x	y	x	y
EKF	1.9042	1.9836	0.1045	0.2788	1.0844	1.0605
UKF	0.4113	1.1985	0.0642	0.1266	0.4825	0.9536
(Improved)	(78.40%)	(39.58%)	(38.56%)	(54.59%)	(55.51%)	(10.08%)
CKF	0.6523	1.3720	0.0857	0.1382	0.5028	0.9783
(Improved)	(65.74%)	(30.83%)	(17.99%)	(50.43%)	(53.63%)	(7.75%)
REKF	0.7566	0.7721	0.0638	0.0625	0.3927	0.9391
(Improved)	(60.27%)	(61.08%)	(38.94%)	(77.58%)	(63.79%)	(11.45%)
SOKF	0.8428	1.7616	0.0793	0.2597	0.7113	1.0198
(Improved)	(55.74%)	(11.19%)	(24.11%)	(6.85%)	(34.41%)	(3.84%)
SORKF	0.2473	0.4087	0.0480	0.0597	0.0457	0.1956
(Improved)	(87.01%)	(78.98%)	(54.07%)	(78.59%)	(95.79%)	(81.56%)

**Table 2 sensors-22-05590-t002:** Truncation error of trajectory estimation in fixed-formation multi-robot.

R	Truncation Error of R1		Truncation Error of R2		Truncation Error of R3	
Method	x	y	x	y	x	y
EKF	1.9536	1.9243	0.1085	0.3117	1.0726	1.0734
SOKF	0.8729	1.7157	0.0905	0.2391	0.6910	0.9819
(Reduced)	(55.32%)	(10.84%)	(16.59%)	(23.29%)	(35.58%)	(8.52%)

**Table 3 sensors-22-05590-t003:** Rounding error of trajectory estimation in fixed-formation multi-robot swarm.

R	Rounding Error of R1		Rounding Error of R2		Rounding Error of R3	
Method	x	y	x	y	x	y
REKF	0.0818	0.0847	0.0179	0.0138	0.0445	0.1845
SORKF	0.0333	0.0487	0.0142	0.0125	0.0130	0.0320
(Reduced)	(59.29%)	(10.84%)	(20.67%)	(9.42%)	(70.79%)	(82.66%)

**Table 4 sensors-22-05590-t004:** MSE for multi-robot trajectory estimation in random topology formation.

R	MSE of R1		MSE of R2		MSE of R3	
Method	x	y	x	y	x	y
EKF	0.0458	0.0886	0.1895	0.9687	0.1523	0.3874
UKF	0.0431	0.0632	0.1382	0.4635	0.0757	0.0877
(Improved)	(5.90%)	(28.67%)	(27.07%)	(52.15%)	(50.29%)	(77.36%)
CKF	0.0362	0.0649	0.0825	0.2875	0.0586	0.0732
(Improved)	(20.96%)	(26.75%)	(56.46%)	(70.51%)	(61.52%)	(81.10%)
REKF	0.0390	0.0426	0.1391	0.2143	0.0061	0.0338
(Improved)	(14.85%)	(51.92%)	(26.60%)	(77.88%)	(95.99%)	(91.28%)
SOKF	0.0433	0.0451	0.1497	0.4809	0.0412	0.0850
(Improved)	(5.46%)	(49.10%)	(21.00%)	(50.36%)	(72.95%)	(66.75%)
SORKF	0.0340	0.0366	0.0191	0.0433	0.0044	0.0325
(Improved)	(25.76%)	(62.08%)	(89.92%)	(95.53%)	(97.11%)	(91.61%)

**Table 5 sensors-22-05590-t005:** Truncation error of trajectory estimation in random topology formation.

R	Truncation Error of R1		Truncation Error of R2		Truncation Error of R3	
Method	x	y	x	y	x	y
EKF	0.0629	0.0905	0.2299	1.0499	0.1919	0.4147
SOKF	0.0408	0.0492	0.1579	0.4622	0.0606	0.1625
(Reduced)	(35.14%)	(45.64%)	(29.16%)	(55.98%)	(68.42%)	(60.82%)

**Table 6 sensors-22-05590-t006:** Rounding error of trajectory estimation in random topology formation.

R	Rounding Error of R1		Rounding Error of R2		Rounding Error of R3	
Method	x	y	x	y	x	y
REKF	0.0165	0.0136	0.0543	0.0861	0.0124	0.0235
SORKF	0.0126	0.0121	0.0111	0.0110	0.0117	0.0169
(Reduced)	(23.64%)	(11.03%)	(79.56%)	(87.22%)	(5.65%)	(28.09%)

## Data Availability

Not applicable.

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
