# Peer review of "A High-Order Kalman Filter Method for Fusion Estimation of Motion Trajectories of Multi-Robot Formation"

_sensors, 2022, doi:10.3390/s22155590_

Round 1
Reviewer 1 Report
This paper proposes a multi-robot formation trajectory based on the high-order Kalman filter method. The main weaknesses that can be improved:
1) As shown in Table 4 in Section 5.2, why are all the x-axis MSE values smaller than the y-axis MSE values
2) The references are mainly related to IEEE Transactions on Automatic Control and other IEEE Transactions. It seems not completely clear, why the Authors submitted their paper to Sensors instead of the mentioned journals. The Authors are encouraged to add some relevant references to this journal, demonstrating fitting the paper to the interests of the journal readers?
3) There are some typos that need to be carefully checked.
4) There are some grammar mistakes. The author should check carefully.
Author Response
We thank four reviewers for their suggestions. These comments are valuable and helpful to our paper. We have read all these suggestions carefully and made corrections and improvements one by one. Based on the instructions, we upload the revised manuscript file. We hope the new version is smoother and clearer. For the specific reviewer's responses, we present in following.

Reviewer 2 Report
This paper presents a dynamical state estimation algorithm based on non-linear Kalman filtering with N-order Taylor approximation which can be also applied in the context of multi-robot formation state estimation problems.
With respect to EKF (Extended Kalman Filter) which relies on a first-order Taylor expansion, performance are increased. Remainder variables taking into account the truncation error can be included in the state model.
The paper is interesting and the algebraic formulation is detailed. However, some relevant aspects should be discussed.
1) A comparison with UKF and CKF is missing in the paper. It would be important to show the method performance with respect to other state-of-the-art algorithms such as UKF and CKF applied to the same simulation environment. Processing time and algorithm complexity must be discussed, too.
2) my doubt concerns whether by filling the transition matrix with all contributions from the various n-hexima derivatives uses more computational power than a UKF. Authors may want to discuss this possible outcome.
3) I would suggest representing the localization errors in more immediate modes such as histograms, cdf or boxplot.
4) English must be improved. There are some sentences presenting a not allowed structure, e.g. "Time domain filtering method: Kalman filter (KF).". Several english flaws are present.
5) What do authors intend with: "UKF does not need to calculate the Jacobian matrix, and the accuracy can reach at least order 2." Please, clarify what an "order 2" is reffered to.
6) Other high-order extended kalman filters are present in literature and are also cited in your manuscript (e.g. 9 reference 9 and 10). What are the advantages of the proposed algorithm?
Author Response

(The authors gave the same response as above.)

Reviewer 3 Report
The authors proposed a multi-robot formation trajectory based on the high-order Kalman filter method. The paper was well written and well organized. Therefore, I have a few queries as follows:
- The abstract is excessively straightforward. The authors should add some background information regarding this work to grab the reader's attention.
- The work is well written. However, the author must add more references. The paper has 20 pages and only 14 references. Also, they should add more recent references. It would improve the introduction, solidify the comparison with other work, and stress the contribution of the work.
- Finally, as a suggestion, it would be an improvement if the authors present in the conclusion section some quantitative results (percentage or other metrics). For instance: what was the improvement relative to the compared work, how much was the error reduced, and how much was the gain in processing time.
Author Response

(The authors gave the same response as above.)

Reviewer 4 Report
The manuscript is drafted well and has provided acceptable results and observations.
I would recommend these incorporations
1. In Abstract: The author/s have directly jumped to Kalman filters, provided the application and research gap could have been highlighted in first sentence.
2. Let the author add accuracy and results outcomes in a measurable format for ease to the readers
3. Add the following literature
Basha, Syed Muzamil, Syed Thouheed Ahmed, and Naif K. Al-Shammari. "A Study on Evaluating the Performance of Robot Motion Using Gradient Generalized Artificial Potential Fields with Obstacles." In Computational Intelligence in Data Mining, pp. 113-125. Springer, Singapore, 2022.
Dean, G. C. "An introduction to Kalman filters." Measurement and Control 19, no. 2 (1986): 69-73.
Lee, T. Sen. "Theory and application of adaptive fading memory Kalman filters." IEEE transactions on circuits and systems 35, no. 4 (1988): 474-477.
4. The author needs to add "Block/Architecture" diagram in methodology of proposed technique
5. Add a table to represent mathematical symbols
Author Response

(The authors gave the same response as above.)

Round 2
Reviewer 2 Report
About Response #3: In my opinion, both punctual localization error and CDF/histograms should be represented to better show the performance of the various methods analysed. The usage of more than 50 points for the computation of CDF/histograms must be considered. Alternatively, Monte Carlo analysis can be used.
About Response #5: authors write:
"The computational effort of the UKF algorithm is the same as that of the EKF, and the filtering accuracy is better than that of the EKF, which can at least reach the second-order accuracy of Taylor expansion"
Please, check this affermation. UKF applies some data processing (creation of sigma points, propagation of sigma points) which surely increase the computational burden with respect to EKF.
Author Response
We thank the reviewer for suggestions. These comments are valuable and helpful to our paper. We have read all these suggestions carefully and made corrections and improvements one by one. Based on the instructions, we upload the revised manuscript file. We hope the new version is smoother and clearer. For the specific reviewer's responses, we present in following.
